# ATTIRE: Albumin To prevenT Infection in chronic liveR failurE: study protocol for an interventional randomised controlled trial

Louise China,[1] Simon S Skene,[2] Kate Bennett,[3] Zainib Shabir,[3] Roseanna Hamilton,[3] Scott Bevan,[3] Torsten Chandler,[3] Alexander A Maini,[1] Natalia Becares,[1] Derek Gilroy,[1] Ewan H Forrest,[4] Alastair O'Brien[1]

¹Division of Medicine, University College London, London, UK
²School of Biosciences and Medicine, University of Surrey, Guildford, UK
³Comprehensive Clinical Trials Unit, UCL, London, UK
⁴Department of Gastroenterology, Glasgow Royal Infirmary, NHS Greater Glasgow and Clyde, Glasgow, UK

**Correspondence to**
Dr Louise China;
louise.china@ucl.ac.uk

## ABSTRACT

**Introduction** Circulating prostaglandin $E_2$ levels are elevated in acutely decompensated cirrhosis and have been shown to contribute to immune suppression. Albumin binds to and inactivates this immune-suppressive lipid mediator. Human albumin solution (HAS) could thus be repurposed as an immune-restorative drug in these patients. This is a phase III randomised controlled trial (RCT) to verify whether targeting a serum albumin level of ≥35 g/L in hospitalised patients with decompensated cirrhosis using repeated intravenous infusions of 20% HAS will reduce incidence of infection, renal dysfunction and mortality for the treatment period (maximum 14 days or discharge if <14 days) compared with standard medical care.

**Methods and analysis** Albumin To prevenT Infection in chronic liveR failurE stage 2 is a multicentre, open-label, interventional RCT. Patients with decompensated cirrhosis admitted to the hospital with a serum albumin of <30 g/L are eligible, subject to exclusion criteria. Patients randomised to intravenous HAS will have this administered, according to serum albumin levels, for up to 14 days or discharge. The infusion protocol aims to increase serum albumin to near-normal levels. The composite primary endpoint is: new infection, renal dysfunction or mortality within the trial treatment period. Secondary endpoints include mortality at up to 6 months, incidence of other organ failures, cost-effectiveness and quality of life outcomes and time to liver transplant. The trial will recruit 866 patients at more than 30 sites across the UK.

**Ethics and dissemination** Research ethics approval was given by the London-Brent research ethics committee (ref: 15/LO/0104). The clinical trials authorisation was issued by the medicines and healthcare products regulatory agency (ref: 20363/0350/001–0001). The trial is registered with the European Medicines Agency (EudraCT 2014-002300-24) and has been adopted by the National Institute for Health Research (ISRCTN 14174793). This manuscript refers to version 6.0 of the protocol. Results will be disseminated through peer-reviewed journals and international conferences. Recruitment of the first participant occurred on 25 January 2016.

## Strengths and limitations of this study

► This is the first prospective, interventional randomised controlled trial with the primary aim of assessing the effects of serum albumin targeted 20% human albumin solution (HAS) infusions on infection, organ failure and death in hospitalised patients with decompensation of liver cirrhosis.

► Study design was optimised by a prior 80-patient feasibility study.

► Cost-effectiveness and quality of life assessment will review whether any outcomes are sustainable and feasible in a national health service setting.

► The study is unblinded to site investigators due to lack of agreement regarding the control infusion fluid and high cost required.

► Patients in the control arm may receive 20% HAS as part of current international guidance as it would be unethical to withhold this in certain situations.

## INTRODUCTION

Liver disease is an increasing cause of mortality currently in the UK and is the fifth most common cause of death.[1] These deaths are predicted to double over the next 20 years[2]. Mortality is mainly seen in the middle-aged group. Liver disease has surpassed lung and breast cancers as the leading causes of years of working life lost, and are set to overtake ischaemic heart disease within 2–3 years[3].

Patients with complications of liver failure secondary to liver cirrhosis are described as decompensated patients. They are highly prone to bacterial infection[4] secondary to immune dysfunction,[5] with nosocomial (hospital-acquired) infection rates of 35% compared with 5% in non-cirrhotic patients.[6 7] Of those who develop infection with organ dysfunction, 60%–95% will die, often following prolonged intensive care unit

(ICU) admission.[8] There is, however, no medical strategy to restore immune competence.

It has been demonstrated that elevated circulating prostaglandin $E_2$ ($PGE_2$) levels contribute to immune suppression in acutely decompensated patients.[9] The plasma protein albumin binds and catalyses the inactivation of $PGE_2$.[10] The binding capacity of endogenous albumin is known to be defective in cirrhosis[11 12] and improves following albumin infusions.[13] Albumin is synthesised in the liver and levels fall as the synthetic function of the liver declines in advanced cirrhosis. Therefore, low albumin levels, with defective binding function, make $PGE_2$ more bioavailable. We found a serum albumin of <30 g/L predicted plasma-induced macrophage dysfunction in a small cohort of decompensated cirrhosis patients[9] and this was reversed when albumin levels were increased to ≥30 g/L. This finding has recently been validated in a multicentre clinical trial[13]; however, it is not known if these ex vivo observations translate into an improvement in clinical outcomes.

We propose a novel strategy to repurpose 20% human albumin solution (HAS) as an immune-restorative drug in patients with decompensated liver cirrhosis with the aim of maintaining serum albumin at near-normal levels.

The Albumin To prevenT Infection in chronic liveR failurE (ATTIRE) phase II single-arm, multicentre feasibility trial[14] (n=80) verified that daily intravenous human albumin infusions restored serum albumin levels to near normal in hospitalised patients with decompensated cirrhosis, that it was safe and that there was physician equipoise prior to proceeding to a large randomised controlled trial (RCT).[15] It has also allowed refinement of our definitions of RCT outcomes.

This phase III RCT (n=866) will assess the impact of 20% HAS treatment on the incidence of nosocomial infections, organ dysfunction and mortality in patients admitted to hospital with decompensation of liver cirrhosis.

## METHODS AND ANALYSIS
### Primary objective

To verify whether raising and maintaining serum albumin levels ≥30 g/L in patients admitted with decompensated cirrhosis using repeated 20% HAS infusions will reduce incidence of infection, renal dysfunction and mortality for the treatment period (maximum 14 days or discharge if <14 days) compared with standard medical care (figure 1). To raise and maintain serum albumin levels above 30 g/L, 20% HAS will be prescribed and administered in the intervention arm if levels are below 35 g/L.

Secondary objectives are to assess the impact of the intervention on development of other organ dysfunction (cardiac, respiratory, brain and liver), cost-effectiveness, quality of life (QOL) and sustainability of any outcomes over a 6-month follow-up period.

### Trial design

This is a multicentre, open-label, RCT in which patients will either be treated with 20% HAS to raise and maintain serum albumin above 30 g/L or continue to receive their usual standard of care treatment. Eight hundred sixty-six sequential patients admitted to 30 UK participating hospitals with a clinical diagnosis of cirrhosis and decompensation will be screened using the inclusion and exclusion criteria (table 1). Decompensation includes: jaundice, ascites, hepatic encephalopathy, variceal bleeding, coagulopathy and hepatorenal syndrome (HRS).

As advanced liver disease requires frequent hospital readmissions, patients may be enrolled in the RCT more than once, with a 30-day 'washout period' following discharge after the first enrolment, during which time they cannot be enrolled for a second time. This is to account for albumin's half-life of 18–21 days[16]. Patients re-entering the trial in this way will be rerandomised, so that each enrolment will be considered as an independent patient 'presentation'.[17]

To ensure homogeneity in the approach to patient recruitment, intervention and data collection, all sites received introduction training and regular follow-up re-training plus monitoring and support visits from the sponsor.

### Clinical trial endpoints
#### Primary endpoint

A composite endpoint comprising incidence of infection, renal dysfunction and mortality within the treatment period (for a maximum of 14 days OR when the patient is considered fit for discharge if <14 days).

The three components of the composite endpoint are:
1. New infection: indicated by clinician diagnosis and clinical evidence provided on completed infection case report form (CRF). Blinded data from a clinically representative sample of patients will be scrutinised for the presence of infection by a clinical trial endpoint review committee. The committee will contain microbiologists who will validate the presence of infection according to peer reviewed criteria[18] (see online supplementary appendix 1).
2. Renal dysfunction: a serum creatinine increase of ≥50% as compared with serum creatinine at randomisation OR the patient is initiated on renal replacement support (either haemodialysis or haemofiltration) OR a rise in serum creatinine of ≥26.5 µmol/L within 48 hours.
3. Mortality

Primary endpoint data will be recorded throughout the treatment period; however, only contributing events captured on the treatment day 3 CRF through to the day 15 CRF (the end of treatment examination day) will contribute to the primary outcome. This allows the biological effect of albumin to be established at least 24 hours following initiation of treatment.

If the participant is discharged or deemed medically fit for discharge prior to day 15, no further primary outcome

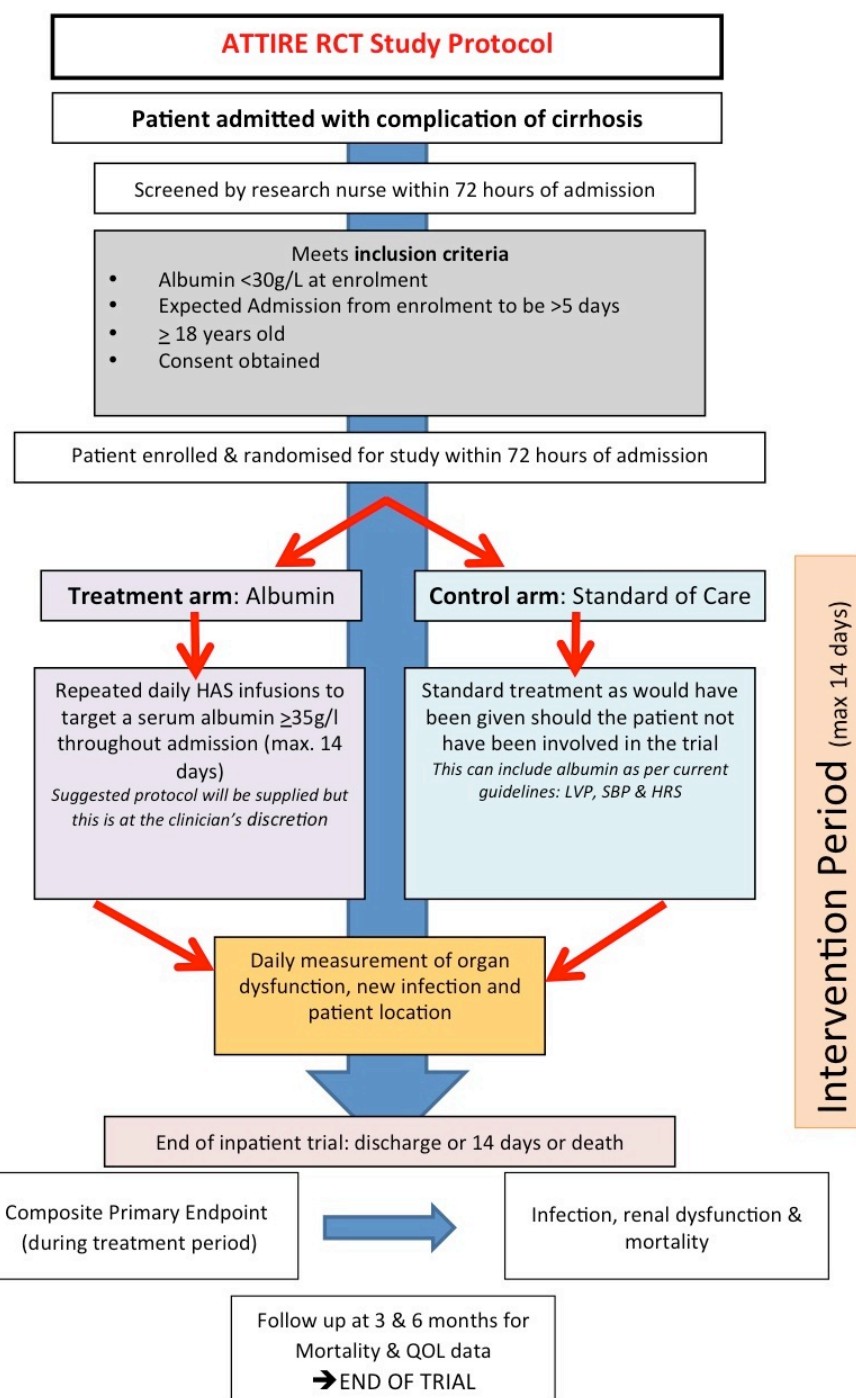

**Figure 1** Overview of Albumin To prevenT Infection in chronic liveR failurE (ATTIRE) randomised controlled trial (RCT) study protocol. HAS, human albumin solution; HRS, hepatorenal syndrome; LVP, large volume paracentesis; QOL, quality of life; SBP, spontaneous bacterial peritonitis.

data will be measured after this date, as this will signal the end of the participant's treatment period.

### Secondary endpoints

1. Mortality at 28 days postrandomisation and at 3 and 6 months postdischarge
2. Time to outcome (first event of infection/renal dysfunction/death)
3. Incidence of respiratory, circulatory and cerebral dysfunction during the treatment period (based on modified components of the Chronic liver failure-sequential organ failure assessment (CLIF-SOFA)[19] score).
4. Incidence of liver transplant within 6 months of treatment
5. Total amount of HAS administered during the treatment period
6. Duration of hospital stay
7. Change in prognostic score (assessed by Model for End Stage Liver Disease (MELD)) between randomisation and end of treatment period.
8. Days in ICU during treatment period

**Table 1** Patient selection criteria

| Patient inclusion criteria | Patient exclusion criteria |
| --- | --- |
| All patients admitted to hospital with acute onset or worsening of complications of cirrhosis | Advanced hepatocellular carcinoma with life expectancy of less than 8 weeks |
| Over 18 years of age | Patients who will receive palliative treatment only during their hospital admission |
| Predicted hospital admission ≥5 days at trial enrolment, which must be within 72 hours of admission | Patients who are pregnant |
| Serum albumin <30 g/L at screening | Known or suspected severe cardiac dysfunction |
| Documented informed consent to participate (or consent given by a legal representative) | Any clinical condition which the investigator considers would make the patient unsuitable for the trial |
| | The patient has been involved in a clinical trial of investigational medicinal products (IMPs) within the previous 30 days that would impact on their participation in this study |
| | Trial investigators unable to identify the patient (by NHS number) |

NHS, National Health Service.

9. Incremental cost and cost-effectiveness up to 6 months postdischarge
10. Impact on QOL up to 6 months postdischarge
11. Safety and tolerability of HAS as indicated by serious adverse events
12. Use of terlipressin for (1) renal dysfunction, (2) hypotension and (3) variceal bleeding.

Patients in the RCT will be followed up 3 and 6 months (+/−1 month) following discharge from hospital. Exploratory objectives are listed in online supplementary appendix 2.

### Patient population
This will include all patients admitted to hospital with complications of decompensated liver cirrhosis and serum albumin <30 g/L, aged over 18 years with anticipated hospital length of stay of 5 or more days at trial enrolment, which should be no later than 72 hours from admission. This is subject to exclusion criteria as detailed in table 1. The diagnosis of cirrhosis will be made by the clinical team as per standard UK practice and does not require liver biopsy or imaging. Acute decompensation of liver cirrhosis associated with organ failures is termed ACLF (acute on chronic liver failure). ACLF has a number of definitions[20 21] based on the SOFA score, all are associated with a poor prognosis. This study will include patients with decompensated cirrhosis with and without ACLF; it will also record the development of ACLF during the study treatment period.

### Consent
Patient information sheets (see online supplementary appendix 3) will be given to and discussed with potential patients before consent is sought. Informed consent will be obtained from each participant or his or her legal representative. Patients who lack mental capacity, for any reason, are not excluded from the trial. An important subgroup of patients will have hepatic encephalopathy and these patients may lack capacity to consent; however, these patients may be among those that receive maximum benefit from the intervention.[21–23] In this case, consent will be sought from an appropriate legal representative independent of the research team as per current UK clinical trials regulations.[24]

### Intervention
After randomisation (when serum albumin is <30 g/L) patients will receive either daily dose of 20% HAS intravenously if their serum albumin level is less than 35 g/L (at approximately 100 mL/hour) or standard medical care (which may include 20% HAS infusions for indications listed in established guidelines only; see below) for a maximum of 14 days or discharge (if <14 days). The volume of HAS each day will be determined by the patient's serum albumin level on that day (or the closest previous measurement if there are no results from that day available).

Table 2 shows the suggested dosing protocol for albumin administration in the treatment arm group. This is a suggested regimen and responsible clinicians are given flexibility to alter this depending on the clinical situation. The effectiveness of this protocol, and approach, was verified in the ATTIRE feasibility study.[15] Differing regimens may be used to cover large volume paracentesis (LVP) (8 g of albumin per litre of ascites drained) or treat HRS (1 g of albumin per kilogram of body weight) as per international guidelines[25 26] but HAS *must* be prescribed and given if serum albumin <35 g/L unless there are any safety concerns. All variations will be recorded in the patient's daily CRF.

Twenty per cent HAS will only be given in the standard of care arm if the patient requires large volume paracentesis

**Table 2** Treatment arm dosing protocol for 20% human albumin solution (HAS) administration (amounts per day) as advised by measured serum albumin level on that day

| Patient's serum albumin level | Amount of 20% HAS to be administered |
| --- | --- |
| ≥35 g/L | none |
| 30–34 g/L | 100 mL |
| 26–29 g/L | 200 mL |
| 20–25 g/L | 300 mL |
| <20 g/L | 400 mL |

or has spontaneous bacterial peritonitis (SBP) or HRS (as per established guidelines[26–28]). This needs to be clearly recorded in the patient's CRF, and if HAS is given for any other indication in the Standard of Care arm, this will be considered a protocol deviation. The administration of HAS in the Standard of Care arm will be closely monitored by the Independent Data Monitoring Committee (IDMC).

Randomisation will use a minimisation algorithm incorporating a random element, stratifying by centre, MELD score, and number of organ dysfunctions, serum albumin level and if antibiotics are currently being prescribed. To ensure maximum balance is achieved across the stratification factors, minimisation will be carried out on these factors separately.

### Evaluations during and after treatment

Clinical, biochemical, microbiological, health economic and QOL data will be collected during the trial treatment period (see online supplementary appendix 4) using information from hospital notes that is recorded as standard of care. There is follow-up at 3 and 6 months. The blood, urine and stool samples collected for immune function tests will be analysed in a blinded fashion at a central site.

### Patient and public involvement

Patients who have had a previous diagnosis of liver cirrhosis were involved in the development of the research question and outcome measures. Patient feedback led to the addition of QOL evaluation. The study protocol and patient information leaflets were reviewed by our patient representatives with positive feedback. In addition, an independent patient representative sat on the ethics approval panel. Results will be disseminated to patients via open access publication and our local trials teams.

### Statistical considerations
#### Sample size

A 30% incidence of nosocomial infection in acute decompensated cirrhosis patients is well documented with up to 30% of these patients developing organ dysfunction[21 29] and an overall mortality of 38% at 1 month.[18 21 29] These figures are supportive of 30% as a conservative estimate for the composite primary endpoint of incidence of new infection, renal dysfunction or mortality up to 14 days from randomisation.

We have assumed that the 'immune-restorative' albumin treatment would reduce this rate by 30% to a rate of 21%, which would be considered clinically relevant. Three hundred eighty-nine patients per arm would be sufficient to detect such a difference with 80% power at a significance level of 0.05. Allowing for loss to follow-up/withdrawal of 10% from the trial, we will aim to recruit 433 to each arm (866 in total).

### Statistical evaluation

A detailed statistical analysis plan will be written and approved by the Trial Steering Committee (TSC) before the substantive analysis of unblinded trial data.

Baseline characteristics will be summarised by treatment using appropriate descriptive statistics, means and SD for approximately normally distributed variables, medians and IQRs for non-normally distributed variables and counts (percentages) for categorical variables.

#### Primary outcome

The primary outcome is the difference in event rates, according to treatment, of the composite endpoint of infection, renal dysfunction and mortality within the intervention period (from ≥24 hours from the start of treatment/randomisation up to a maximum of 14 days or up to discharge if this is prior to 14 days).

Since the primary outcome has a binary classification, logistic regression will be used to determine whether there is any difference in rates due to treatment, by inclusion of a binary covariate indicating treatment. The results will be adjusted for predetermined prognostic factors used as stratifying variables in the randomisation, which shall be included as additional covariates in the model. The model coefficient due to treatment will give an estimate of the difference in log odds, or equivalently (exponentiated to give) an estimate of the odds multiplier, that is, the change in odds of a negative outcome on the composite endpoint due to treatment with albumin. It is expected that this effect will be negative, so that treatment with albumin is seen to reduce the odds of a negative outcome. A reduction from 30% to 21% would be associated with a reduction in odds of around 38%. Predicted probabilities will be presented for the composite outcome for each of the treatment arms, adjusted for the model covariates.

The effect of treatment on the individual components, infection, renal dysfunction and mortality will be reported separately.

#### Secondary and exploratory outcomes

Secondary and exploratory outcomes will be handled similarly. Continuous secondary outcome measures will be analysed using a linear regression model, and time-to-event outcomes using Kaplan-Meier plots and Cox's proportional hazards models where covariate adjustment is necessary.

All statistical tests will use a two-sided p value of 0.05, unless otherwise specified, and all CIs presented will be 95% and two-sided. All statistical analyses will be performed using Stata (StataCorp, College Station, Texas, USA).

### Economic evaluation

The economic evaluation will consist of two parts. The primary analysis will be a decision analytic model comparing the expected lifetime costs and benefits of treatment reported in terms of the cost per incremental quality-adjusted life-years (QALYs) gained. QALYs will

be calculated based on the health-related QOL (HRQL) and mortality data collected during the trial. HRQL will be measured according to the 5 level EQ-5D (EQ-5D-5L) (www.euroqol.org), which we will collect at baseline, discharge, 3 months and 6 months (+/−1 month) for each individual patient. If patients are rerandomised, they will have separate follow-up events for each episode of rerandomisation, unless the patient is rerandomised within the follow-up phase of the study. Utility scores will be calculated using UK-specific tariffs and adjusting for baseline differences in patients in the trial arms if necessary. These will be extrapolated over the expected lifetime of the patient and where necessary will draw on the existing literature. The final form of the decision analytic model will be determined during the trial period. Appropriate methods for estimating changes in health over time, such as state-transition models, will be given consideration.

The secondary economic analysis will be a 'within-trial' analysis, reporting on the incremental cost and benefits observed during the trial period and follow-up periods. The cost-effectiveness measures in the within-trial analysis will be the incremental cost per infection avoided and the incremental cost per QALY gained. Costs will be measured as described below. Infections avoided will be derived from the primary outcome in the trial.

Patient resource use will be assessed using hospital patient records and a self-complete resource use form, the client services receipt inventory (CSRI). The volume of resource use for each cost component will be measured directly in the trial from patient records and using the CSRI. A detailed economic evaluation analysis plan will be developed for approval by the TSC.

## DISCUSSION

ATTIRE is a UK multicentre trial that aims to evaluate the repurposing of HAS as an immune-restorative drug. This protocol describes the RCT which will determine if administering daily 20% HAS infusions in order to increase serum albumin to near-normal levels after admission to hospital will reduce rates of infection, renal failure and death in patients with decompensation of liver cirrhosis.

In liver cirrhosis, current evidence-based guidance[26–28] advocates the use of HAS in large-volume paracentesis,[30] HRS[31] and SBP.[32 33] The mechanism behind current usage of albumin is that of volume expansion. However, many hepatologists believe albumin has other medicinal properties; however, a large-scale interventional clinical trial is yet to support this hypothesis. Ex vivo analysis from the ATTIRE feasibility study supported an immune-restorative role for albumin[13]; however, it is unknown whether this correlates with clinical outcome. Our inclusion criteria are broad and designed to represent the decompensated cirrhotic population admitted to UK hospitals in order to make trial outcomes highly relevant in an era of soaring liver disease with few available treatment options. Preventing a liver cirrhosis-related infection

could potentially provide a bridge to liver transplantation, which represents the only curative option for these patients.

The design of this RCT has been refined based on the outcomes of an 80-patient interventional feasibility study.[15] Our targeted infusion protocol (table 2) was effective at increasing serum albumin, even in patients with a very low starting serum albumin. Many hepatology trials focus solely on 28-day survival; however, mortality requires large sample size and may be caused by other factors unrelated to infection. We considered that endpoints for interventions to prevent infection should be assessed at an earlier stage during the patient admission using ward-based clinical measures and laboratory data. Multiple components of a primary composite endpoint were tested in our feasibility study.[15] Defining infection according to clinician diagnosis, triggering completion of an infection CRF, with a substantial proportion blindly scrutinised by a microbiology panel worked well in our feasibility study. Patients that developed renal dysfunction as assessed by an increase in serum creatinine had a poor prognosis, as expected. However, only one patient developed cerebral dysfunction reflecting severe hepatic encephalopathy (>grade 3) suggesting under-reporting and the majority of patients that solely triggered respiratory and cardiovascular endpoints had a good outcome. This is counterintuitive as organ dysfunction is a key predictor of poor prognosis and we consider our data cast significant doubt over whether these dysfunctions can be recorded accurately in largely ward-based patients across multiple sites. This precludes their use as part of our RCT primary composite endpoint; although these will be reported.

Studies evaluating the safety of HAS infusions have generally shown it to be a safe treatment.[30 34–36] The main concerns in the cirrhotic population are related to volume overload leading to pulmonary oedema and increase in portal pressure leading to variceal bleeding. A recent interventional trial in septic decompensated patients reported an 8.3% rate of pulmonary oedema in the albumin treatment group.[37] However, the weight-based albumin dosing regimen on days 1 and 3 in this study led to much larger volumes of albumin prescribed than suggested in our protocol for these days. This and other studies in cirrhosis have not reported an increased incidence in variceal bleeding.[30 33 38]

A challenge this study faces is the lack of a comparator infusion fluid in the control arm and hence the impossibility of site investigator and patient blinding. We had therefore considered comparison with another fluid such as 0.9% sodium chloride; however, this posed a number of problems. Physician preference for resuscitation fluid varies substantially and, in preliminary discussions, many objected to the additional salt load associated with 0.9% sodium chloride, in the non-albumin arm, if the patient did not require fluid resuscitation. Administration of 20% HAS for SBP, LVP and HRS is standard of care, and therefore, if patients in either arm developed these complications during their admission, they would

have to be unblinded to see if they were already receiving 20% HAS as part of the trial. As a significant number of patients do develop these complications, the consequent number of unblinded patients could have rendered the process invalid. Therefore, the RCT had to be an open-label study. This risks biasing adverse event reporting in the treatment arm, making albumin appear unsafe. We therefore felt that an open-label trial design versus a comparator arm of current standard of care was the only pragmatic solution. It could be argued that any improved outcome in the albumin treatment arm may simply be due to additional fluid resuscitation. The mean volume of 20% HAS infused over the treatment period (14 days) was 1 L in our feasibility study.

The regulation of HAS administration in the standard of care (control) arm poses a challenge as HAS is routinely used in clinical practice based on evidence-based guidelines.[26–28] The protocol allows use for these indications. If HAS is administered to control arm patients outside of current evidence-based guidance, this will be defined as a protocol deviation and the IDMC will be monitoring these events closely.

The ATTIRE RCT (n=866, up to 40 centres) began recruitment in April 2016 and aims to complete in December 2019. The medicinal properties of albumin in the hepatological community have long been questioned. The outcomes of this large, multicentre, RCT will be highly relevant to improving the care of hospitalised patients with decompensation of chronic liver disease.

**Acknowledgements** We have had the support of the following individuals via trial oversight committees: Professor Graeme Alexander, Professor Stephen Brett, Professor Mauro Bernardi, Professor Dominique Valla, Dr Vipul Jaraith, Mr Tim Clayton, Mr Brennan Kahan, Dr James O'Beirne, Dr Jim Portal, Dr Gavin Wright, Dr Ewan Forrest, Dr Stephen Ryder, Dr Yiannis Kallis, Dr Shahid Khan, Mr John Crookenden (patient representative), Mr Antonie Bakhuijsen (patient representative) and Ms Susan Tebbs. A list of trial sites and PIs can be obtained by contacting attire@ucl.ac.uk

**Contributors** All authors read and approved the final manuscript. LC: protocol development, writing of this manuscript. ZS, RH, SB, TC: protocol development, protocol implementation, manuscript review. AAM, NB, DG: protocol development, scientific oversight. SSS: protocol development, statistical oversight, manuscript review. KB: protocol development, statistical input, manuscript review. EHF: protocol development, manuscript review. AOB: concept and design, protocol development, writing of this manuscript.

**Funding** The work is supported by the Health Innovation Challenge fund (Wellcome Trust and Department of Health) award number 164699. The trial sponsor is UCL with trial management activities conducted by the UCL Comprehensive Clinical Trials Unit.

**Competing interests** None declared.

**Patient consent** Not required.

**Ethics approval** Research ethics positive opinion and approval was given by the London-Brent Research Ethics Committee (ref: 15/LO/0104) which specialise in trials involving patients who lack the capacity to consent.

**Provenance and peer review** Not commissioned; externally peer reviewed.

**Data sharing statement** Primary and secondary outcomes will be published in a peer-reviewed journal as an open access paper. Deindentified participant data will be shared, as part of the trial results. The full study protocol and statistical analysis plans can be requested by emailing attire@ucl.ac.uk.

**Author note** Trial management and monitoring Research Steering Group: The Research Steering Group (RSG) operates on behalf of the funders to ensure that

appropriate milestones have been met in the delivery of the trial. It consists the CI, an independent expert and representatives of the Welcome Trust and Department of Health. Trial Management Group: The Trial Management Group (TMG) comprises the CI, Clinical Research Fellow, Clinical Project Manager, Trial Statistician, Trial Manager, Data Manager, Health Economist and five trial site PIs. The TMG is responsible for developing the design, co-ordination and strategic management of the trial. Trial Steering Committee: The Trial Steering Committee (TSC) is the independent group responsible for oversight of the trial in order to safeguard the interests of trial patients. The TSC provides advice to the CI, CTU, funder and sponsor on all aspects of the trial through its independent chair. Independent Data Monitoring Committee: The Independent Data Monitoring Committee (IDMC) is responsible for safeguarding the interests of trial patients, monitoring the accumulating data and making recommendations to the TSC on whether the trial should continue as planned. It comprises a clinical chair (independent Hepatologist), independent Gastroenterologist and an independent statistician all with expertise in Clinical Trials.

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
