## [Reviewer comments · BMJ Open]

ARTICLE DETAILS

TITLE (PROVISIONAL)	ATTIRE: Albumin To prevenT Infection in chronic liveR failurE: Study Protocol for an Interventional Randomised Controlled Trial
AUTHORS	China, Louise; Skene, Simon; Bennett, Kate; Shabir, Zainib; Hamilton, Roseanna; Bevan, Scott; Chandler, Torsten; Maini, Alex; Becares, Natalia; Gilroy, Derek; Forrest, Ewan; O'Brien, Alastair

VERSION 1 – REVIEW

REVIEWER	Professor FS Larsen Dept. Hepatology, Rigshospitalet, Copenhagen Denmark I am implicated in a study that are looking at long term survival (1 year) in a similar study that is about to be initiated by the EF-CLIF consortium. I am concerned about the lack of definition of the patients that will be included in this study. Please also make at reference to the CLIF-score (a modified SOFA score for liver failure patients) based on the CANONIC data.
REVIEW RETURNED	04-Jun-2018

GENERAL COMMENTS	Please refer to the CLIF published papers that are the based on the mortality score based on the CANONIC study data, and make it clear to the reader if (some) of these patients will suffer from ACLF. Today it may not be so difficult to keep patient alive for 4 or 12 weeks as the medical critical care management has improved significantly. However, the paper will be of potential interest to clinicians.
---

REVIEWER	Alejandro Quiroga-Garza Universidad Autonoma de Nuevo Leon, Facultad de Medicina, Monterrey, Mexico
REVIEW RETURNED	28-Jul-2018

GENERAL COMMENTS	There are a few, minor, grammatical mistakes throughout the manuscript. The study will include patients from more 30 different sites. Perhaps, a description of the physicians'/researchers' training for patient recruitment, treatment, and follow-up can be included to assure homogeneity throughout the study in the different medical centers.
---

VERSION 1 – AUTHOR RESPONSE

Reviewer: 1

Reviewer Name: Professor FS Larsen

Institution and Country: Dept. Hepatology, Rigshospitalet, Copenhagen, Denmark

Please state any competing interests or state 'None declared': I am implicated in a study that are looking at long term survival (1 year) in a similar study that is about to be initiated by the EF-CLIF consortium. I am concerned about the lack of definition of the patients that will be included in this study. Please also make at reference to the CLIF-score (a modified SOFA score for liver failure patients) based on the CANONIC data.

Please leave your comments for the authors below

'Please refer to the CLIF published papers that are the based on the mortality score based on the CANONIC study data, and make it clear to the reader if (some) of these patients will suffer from ACLF.'

We have amended this patient population paragraph (this is in red in the article):

Patient population

This will include all patients admitted to hospital with the complications of decompensated liver cirrhosis and serum albumin < 30 g/L, aged over 18 years with anticipated hospital length of stay of 5 or more days at trial enrolment, which should be no later than 72 hours from admission. This is subject to exclusion criteria as detailed in table 1. The diagnosis of cirrhosis will be made by the clinical team as per standard UK practice and does not require liver biopsy or imaging. Acute decompensation of liver cirrhosis associated with organ failures is termed ACLF (Acute on Chronic Liver Failure). ACLF has a number of definitions(20,21) based on the SOFA (sequential organ failure assessment) score, all are associated with a poor prognosis. This study will include patients with decompensated cirrhosis with and without ACLF, it will also record the development of ACLF during the study treatment period.

and also point 3 of the secondary outcomes (this is in red in the article):

3. Incidence of respiratory, circulatory and cerebral dysfunction during the treatment period (based on modified components of the CLIF-SOFA score).

'Today it may not be so difficult to keep patient alive for 4 or 12 weeks as the medical critical care management has improved significantly. However, the paper will be of potential interest to clinicians'.

We fully agree and have included a 6 month follow up for all patients which includes evaluation to see if the patient has been listed/received a liver transplant as this is likely to be the only curative option for our patient group.

Reviewer: 2

Reviewer Name: Alejandro Quiroga-Garza

Institution and Country: Universidad Autonoma de Nuevo Leon, Facultad de Medicina, Monterrey, Mexico

Please state any competing interests or state 'None declared': None Declared

Please leave your comments for the authors below

'There are a few, minor, grammatical mistakes throughout the manuscript.

The study will include patients from more 30 different sites. Perhaps, a description of the physicians'/researchers' training for patient recruitment, treatment, and follow-up can be included to

assure homogeneity throughout the study in the different medical centers'

Thank you this is very important. We have inserted this text under the heading 'Trial Design' (this is in red in the article):

To ensure homogeneity in the approach to patient recruitment, intervention and data collection all sites received introduction training and regular follow up re-training plus monitoring and support visits from the sponsor.